# A nanoscale view of the origin of boiling and its dynamics

Mirko Gallo [1,2] ✉, Francesco Magaletti[2], Anastasios Georgoulas[2], Marco Marengo [2,3], Joel De Coninck [2] & Carlo Massimo Casciola [1]

In this work, we present a dynamical theory of boiling based on fluctuating hydrodynamics and the diffuse interface approach. The model is able to describe boiling from the stochastic nucleation up to the macroscopic bubble dynamics. It covers, with a modest computational cost, the mesoscale area from nano to micrometers, where most of the controversial observations related to the phenomenon originate. In particular, the role of wettability in the macroscopic observables of boiling is elucidated. In addition, by comparing the ideal case of boiling on ultra-smooth surfaces with a chemically heterogeneous wall, our results will definitively shed light on the puzzling low onset temperatures measured in experiments. Sporadic nanometric spots of hydrophobic wettability will be shown to be enough to trigger the nucleation at low superheat, significantly reducing the temperature of boiling onset, in line with experimental results. The proposed mesoscale approach constitutes the missing link between macroscopic approaches and molecular dynamics simulations and will open a breakthrough pathway toward accurate understanding and prediction.

Boiling is a phenomenon everybody pretends to have a general understanding of. Due to the large latent heat, it is classically used for temperature management in many engineering devices. In fact, dissipating heat is a limiting factor even for novel applications such as quantum computers[1], cryogenic fluids and advanced nano-cooling[2]. However, despite its ubiquity and technological relevance, not a single description is available to date to address the multiscale, non-equilibrium nature of boiling.

Classical theories predict huge wall superheats—namely, the temperature difference between metastable and saturation states—to initiate the bubbles. Actually, bubble formation takes place from a (metastable) fluid persisting in the liquid state at a temperature exceeding liquid/vapor coexistence (saturation temperature). Superheats of only 15–20 °C are found in practice, despite wall temperatures of the order of 300 °C (i.e. superheat of ≃200 °C) are predicted by available theories. This discrepancy is attributed to gas/vapor entrapment within surface imperfections: the so-called cavity model[3,4]. However, recent experiments[5,6] question this interpretation producing

superheat as low as 10 °C when the surface is kept as smooth and homogeneous as possible.

Despite a perfectly smooth surface is hardly achieved and irregularities are most often unavoidable[7], these puzzling results seem to suggest a different origin of the low boiling temperature, hinting at molecular (nanometric) scale processes, where matter granularity manifests itself as thermal noise on top of an overall mesoscopic (continuum) behavior. Confirming/rejecting this conjecture would require a suitable, as yet unavailable, model of out-of-equilibrium nucleation. Describing such an approach and discussing the low boiling onset temperature at scales unreached by experiments on surfaces that are both perfectly smooth and endowed with controlled heterogeneities are indeed the main aims of the present paper.

Nucleation is the incipience of a first-order phase transition, liquid/vapor phase change in particular[8,9]. Focusing on boiling, the temperature increase from a thermodynamically stable state brings the system in metastable conditions with a finite probability of nucleating a bubble. Roughly speaking, thermal fluctuations

[1]Sapienza University of Rome, Rome, Italy. [2]School of Architecture, Technology and Engineering, University of Brighton, Lewes Road, Brighton, UK. [3]Dept. of Civil Engineering and Architecture, University of Pavia, Pavia, Italy. ✉e-mail: mirko.gallo@uniroma1.it

lead to formation of small vapor embryos (nanometric bubbles) surrounded by the liquid mother phase. A rare event consisting of an unlikely concomitance of elementary steps results in a critical cluster (critical bubbles) able to trigger the phase change though the formation of macroscopic bubbles. In the classical theory, the time needed to observe nucleation is related to the free-energy barrier separating metastable and stable basins. Its height, in its turn, strongly depends on temperature and pressure leading to nucleation rates spanning a range of several orders of magnitude, from very fast when approaching spinodal conditions to overwhelmingly slow close to coexistence[10,11]. In these conditions, rare-event techniques are considered the only viable option to address nucleation under small-to-moderate metastability (i.e., large-to moderately large energy barriers)[12]. However, since boiling is a strongly non-equilibrium phenomenon, common free-energy-based techniques should be ruled out. Large deviations theory would be a more appropriate tool[13] but its application to Navier-Stokes dynamics would be a daunting task[14]. In addition to the modeling of out-of-equilibrium nucleation, the coupling with the macroscopic hydrodynamics is also crucial in boiling. In fact, despite the origin of the phenomenon being atomistic (≤1 nm), boiling is an intrinsically multiscale process that also involves macroscopic hydrodynamics and thermodynamics. At this macroscopic level, bubbles and flow strongly interact, showing a rich phenomenology, e.g., transport, latent heat release, coalescence, fragmentation[15,16]. Finally, all the aforementioned processes are influenced by surface adhesion forces, i.e., by the surface wettability.

Given the phenomenological complexity, several mathematical models have been proposed to study single aspects of the problem. However, a holistic and consistent model encompassing the whole range of scales proved very difficult to attain. Hence, all state of art theories lack at least a single, crucial effect associated with a specific scale. For example, macroscopic hydrodynamics neglects nucleation and relies on empirical correlations to introduce the new phase. The ensuing simulations need ad-hoc initial conditions (e.g., pre-existing vapor bubbles) and suitably calibrated, empirical heat/mass transfer models, which severely penalize their predictive capabilities most necessary in view of unexplored novel technologies. Alternatively, non-equilibrium atomistic simulations are currently limited to too small time and spatial scales to be relevant for applications (e.g., for boiling, the required time step of $\Delta t = 0.1$ fs[17] needs 1 million core hours to simulate a sub-micrometrical system for a couple of nanoseconds[18]. On the other hand, on the experimental side, data can be acquired only for length scales exceeding several micrometers, with time resolution limited to milliseconds[19] or their fractions[20], far beyond the range where the process originates and develops.

Clearly, an unexplored area of boiling exists from nucleation up to the inertial scales where nano/microbubbles are crucial. Here we propose and demonstrate by simulations that fluctuating hydrodynamics (FH) coupled with diffuse interface thermodynamics can address the boiling phenomenology in its whole range of scales. Throughout the paper we will call microscopic the physical quantities related to the granularity of matter. These quantities are inherited at the mesoscale as effective models, e.g., thermal noise, capillarity, and wettability. Conversely, we will refer to macroscopic fields as those inherent in the hydrodynamics of the system.

Concerning nucleation, the methodology was already shown particularly robust and able to describe vapor formation in water[10], replicating the cavitation pressure measured in experiments[21,22] across the whole temperature range of the liquid state. In addition, it describes the field fluctuations in precise agreement with experimental measurements[23,24]. Finally, being based on the extension of a macroscopic fluid model à la Navier-Stokes augmented with capillarity, it reproduces all the well-established large-scale multiphase features[25-27].

In silico experiments along this line elucidate the role of wettability both on nucleation and macroscopic bubble evolution. Comparing boiling on smooth walls and on defects of different wettability will allow shedding light on the controversial issue of the low boiling onset temperature on putatively ultra-smooth surfaces. Moreover, sparse hydrophobic, nanometric patches are found sufficient to trigger nucleation at even lower superheat, significantly reducing the onset temperature, in line with experimental results[5,6]. Finally, we provide an effective model based on the classical nucleation theory and accounting for the Tolman length correction to the surface tension to estimate the nucleation temperature.

## Results

### How surface wettability influences boiling

Typically, boiling occurs in fluids when heated by external energy sources. This scenario inspired our in silico experiments. Under these conditions, the liquid, starting from a thermodynamically stable state, becomes progressively metastable by experiencing an isobaric transformation at increasing temperature. In the metastable state, vapor bubble nucleation can occur with a certain probability, depending on the degree of metastability of the system and on the wall wettability. The Fluctuating Diffuse Interface approach, described in the section Methods, enabled the full-scale description of the vapor formation, here achieved with a spatial/temporal resolution, from the nucleation up to the fully developed bubble macroscopic motion. With a modest computational cost (less than one millionths of what a molecular dynamics approach would need), the still unexplored range of spatial scales from few nanometers to microns is investigated, over time intervals ranging from picoseconds to microseconds. In Fig. 1a, b several snapshots of two numerical simulations are reported with time increasing from bottom to top. These two numerical experiments concern ultra-smooth walls with uniform hydrophilic/hydrophobic wettabilities, with Young contact angles $\phi = 30°$ (Fig. 1a) and $\phi = 105°$ (Fig. 1b), respectively. All the quantities are dimensionless as reported in Section Methods. The computational cubic box of side $L = 720$, corresponding to 0.532 µm, is discretized with cubic cells. Overall, the phase transition process can be divided into three different stages. The first one concerns the nucleation phase, where, after an incubation time, supercritical nuclei start forming due to thermal fluctuations (the first two snapshots from the bottom of both columns). The vapor nanobubbles then start expanding with a process that is strongly catalyzed by bubble coalescence events (third and fourth snapshots). The phase transformation is completed when a nano-film of vapor is formed. The observable that quantifies the advancement of boiling is the reduced temperature $\theta = (T - T_{sat})/(T_{spin} - T_{sat})$, with $T$ the fluid temperature, $T_{sat}$ and $T_{spin}$ the saturation and spinodal temperature, respectively. For a given time window, when $\theta \to 0$ the probability to nucleate vapor is zero, and when $\theta \to 1$ the probability to form vapor is one (spinodal decomposition). In the present case $T_{sat} = 0.9500 = 614.65$ K and $T_{spin} = 0.9625 = 622.74$ K.

The time evolution of the spatially averaged reduced temperature at the wall $\langle\theta\rangle_{wall} = 1/|A_w| \int_{A_w} \theta(x,y,0,t)dxdy$, with $|A_w|$ the area of the planar solid wall at $z = 0$, is reported in Fig. 1c. It is worth mentioning that data are averaged over an area of the order of 1. µm², making the statistics very robust with error bars inappreciable on the scale of the graph. The different curves correspond to different wettabilities, as described in the legend. During the first stage, where nucleation is still to occur ($t < 1.0 \times 10^3 = 2.21$ ns), the wall temperature increases independently of the wettability (heat conduction mode). As soon as nucleation takes place, the available heat flux is used to provide the latent heat of vaporization, inducing surface cooling.

The boiling onset temperature $\langle\theta\rangle^\star_{wall}$, defined as the maximum of $\langle\theta\rangle_{wall}$, is found to be progressively shifted to later times at increasing wettability. In fact, recalling that nucleation is a random process, the temperature history shows dependency on the surface

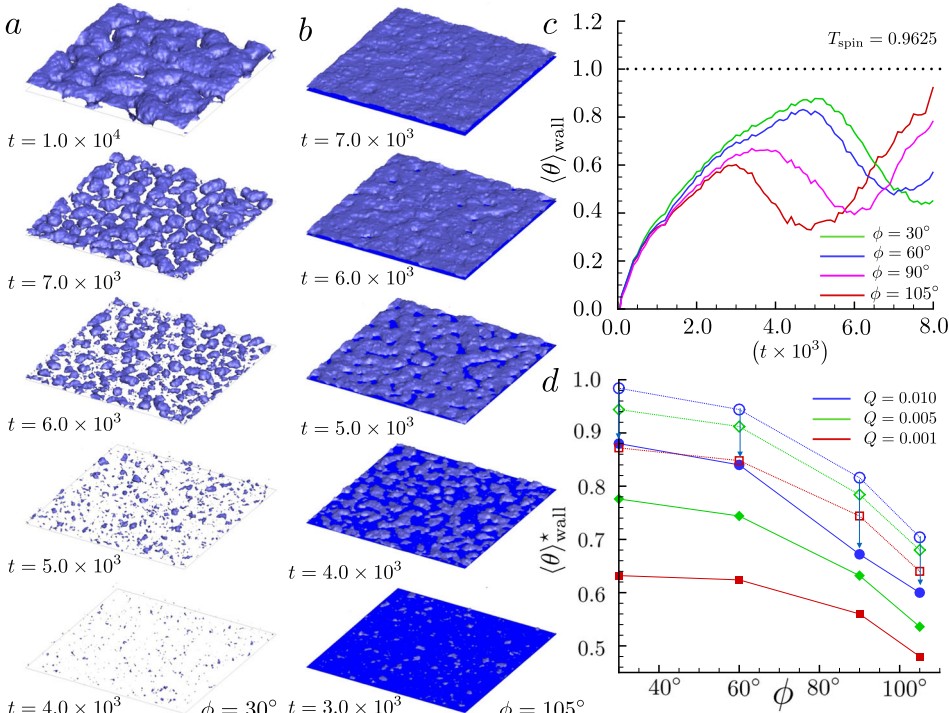

**Fig. 1 | Nucleation on homogeneous wall with different wettability. a** Ultra-smooth wall with a uniform hydrophilic wettability with $\phi = 30°$, the time instant is reported on the left bottom side of the surface. **b** Ultra-smooth wall with a uniform hydrophobic wettability with $\phi = 105°$, also in the latter case the simulation time is reported. **c** Mean reduced wall temperature $\langle\theta\rangle_{\text{wall}}$ vs time. The different lines correspond to different wettabilities, the first relative maximum value of the function identifies the reduced onset boiling temperatures $\langle\theta\rangle^\star_{\text{wall}}$. **d** Mean reduced onset temperature comparison between simulations (solid curves with solid symbols) and CNT prediction. The dotted lines with empty symbols correspond to the CNT with $\delta = 0$. The vertical arrows report the onset temperature theoretical predictions exactly on the numerical values with a suitable choice of the Tolman lengths.

chemistry even before onset conditions as soon as the first vapor embryos start forming. Actually, the onset temperature correlates to the maximum number of formed bubbles (see SI for details).

The dependence of $\langle\theta\rangle^\star_{\text{wall}}$ on the surface chemistry is depicted in Fig. 1d for different imposed heat fluxes $Q$ where the contact angle's strong influence increases with the applied heat flux. The solid curves with solid symbols depict the numerical values. The sensitivity of the onset temperature to the heat flux is the undoubted signature of the strong non-equilibrium nature of boiling. However, despite the non-equilibriuim nature of the process, it is worth comparing the numerical results with an equilibrium (quasi-static) classical approach, the classical nucleation theory (CNT), see section Methods for details. The dotted lines with empty symbols refer to the CNT without the Tolman correction to the surface tension. In the latter case we got onset temperature values significantly higher than those measured by the simulations, as detected in dedicated experiments[6]. However, as recently discussed[9,11,22], often the CNT misprediction can be ascribed to the curvature dependency of the surface tension at the nanoscale. For this reason, we adopted a Tolman-length-corrected CNT to describe our numerical findings. In this case, the surface tension depends on the local curvature of the liquid/vapor interface. It turns out that for typical values of <10 Å, the adoption of a suitable Tolman correction exactly reproduces the onset temperature of FH simulations. The arrows represent the effects of the correction. For the sake of clarity, they are shown only for the case $Q = 0.01 = 75.6$ MWm$^{-2}$, but the correction works for all cases considered assuming a modest variability of the Tolman length (see Supplementary Table. 1).

During the nucleation process, an incubation time $t_{\text{inc}}$—i.e., the time elapsed until the nucleation of the first bubble, see, e.g., Supplementary Fig. 2—is required to form supercritical vapor embryos in the system. Depending on the imposed heat flux, the incubation time spans time

scales between nanoseconds ($t \simeq 10^3 = 2.21$ ns, $Q = 0.01 = 75.6$ MWm$^{-2}$) to microseconds ($t \simeq 10^6 = 2.21$ μs, $Q = 0.001 = 7.56$ MWm$^{-2}$). In order to make contact with applications, it is worth mentioning that a dimensionless heat flux $Q \simeq 10^{-3}$ corresponds to a few MWm$^{-2}$, the typical heat flux used in experiments[7]. It is also interesting to stress that the larger heat fluxes we explored, despite being significantly high, are not unrealistic given the current trend in microelectronics, where modern chips already require to dissipate order 10 MWm$^{-2}$, with peaks that could be even ten times such nominal values[28]. This leads to an important technological challenge in two-phase thermal management, since these heat fluxes are very hard to attain in nucleate boiling, even in subcooled flow boiling conditions[7].

In Fig. 2, the nucleation rate $J_w$ is reported as a function of the contact angle $\phi$ for different values of $Q$, see ref. [15] for a related analysis in the context of cavitation. $J_w$ is defined as the number of supercritical bubbles nucleated on the wall ($N_B(t)$) per unit time and unit area

$$J_w = \frac{1}{|A_w|}\frac{dN_B}{dt} . \tag{1}$$

Numerically, they have been evaluated through a suitable cluster analysis algorithm[16,29] to count the number of bubbles. After an incubation time the number of bubbles linearly increases to a maximum (steady state production rate). Successively, the wall gets crowded with nanobubbles and coalescence leads the bubble number to drastically decrease, see SI for details. The steady-state nucleation rate occurring during the linear stage of the nucleation process can be evaluated as the normalized time derivative of the bubble number, see Eq. (1). As expected, $J_w$ increases with $\phi$, as could have been suggested by free energy considerations[15]. Please note that the reference value of the nucleation rate is $J_{wR} = 8.26 \times 10^{29}$ s$^{-1}$m$^{-2}$.

To summarize, hydrophobic surfaces consistently show much faster vapor production kinetics than their hydrophilic counterparts. The behavior of boiling onset temperature and nucleation rate highlights the key role of wettability on boiling over ultra-smooth surfaces. Specifically, the onset temperature $\langle\theta\rangle^{\star}_{\text{wall}}$ is found to be a decreasing function of the contact angle as observed in experiments[30]. The strong dependence of the onset temperature on the imposed heat flux $Q$ is also coherent with experiments[7,31–33] and a clear indicator of the non-equilibrium nature of boiling at the nanoscale.

### How defects globally influence the boiling process

The vast majority of applications involve surface heterogeneities of various kinds (geometric asperities, chemical non-uniformities). It is, therefore, worth asking what role a nanoscopic defect (a hydrophobic spot on a large hydrophilic plate, for example) may play in the macroscopic dynamics of the phase transition. This is illustrated in Fig. 3a, which concerns the dynamic nucleation of a bubble originating from a hydrophobic defect. In order to focus on a single nucleation event,

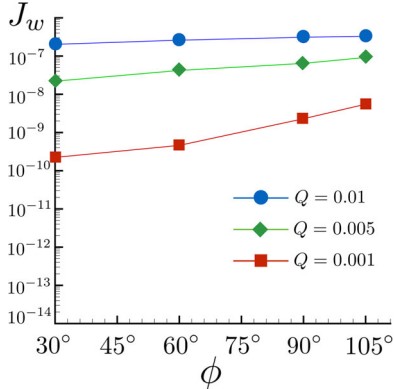

Fig. 2 | **Nucleation rate $J_w$ as a function of the contact angle $\phi$.** The different curves and symbols refer to different values of the external heat flux $Q$.

thus eliminating possible bubble-bubble interactions, we reduced the system size to a nanometric hydrophobic square defect of side $l = 72 -$ equivalent to 53.28 nm – (blue square with $\phi = 105°$) placed in the middle of a hydrophilic wall, $\phi = 30°$, with $L = 240 = 177.6$ n$m$. The initial temperature is $T_{\text{sat}} = 0.900 = 582.3$ K with the spinodal limit as $T_{\text{spin}} = 0.935 = 604.9$ K. Two different, uniform heat fluxes $Q$ are imposed (see caption). The phase change originates somewhere within the hydrophobic defect with the nucleation of a small embryo, which expands in the mother phase. When the bubble is grown enough, the contact line pins to the defect boundaries ($t = 5.4 \times 10^4 = 0.12$ μs). Subsequently, a slipping process allows the bubble to expand over the hydrophilic region. This process requires a certain time to take place, consistently with the notion of thermally activated pinning barrier crossing. Figure 3b shows the time evolution of the mean reduced temperature $\langle\theta\rangle_{\text{wall}}$ (blue solid line) in comparison with the ideal case of a perfectly uniform hydrophilic surface (red dashed curve). For both the values of heat flux, the boiling onset is anticipated by the hydrophobic spot. Interestingly, the effect is more pronounced for the smaller $Q$. In addition, it is worthwhile noting the pronounced temperature drop for boiling occurring in the case of uniform hydrophilic surfaces, where the temperature goes below the saturation value. This feature is not detected when the hydrophobic defect is present on the wall. In the latter case, a single bubble appears on the defect, and results in the entire surface cooling which inhibits bubble formation elsewhere. When homogeneous walls are considered, the nucleation is delayed until it eventually takes place by producing bubbles scattered all over the wall. As a consequence of the larger amount of vapor and related latent heat absorption the temperature drops, leading the mean wall temperature to fall for a while below saturation. Afterwards, the two curves show the same behavior.

The important point here is that just a single defect is shown able to significantly anticipate the nucleation time. Clearly, a single defect is rather artificial. Actually, the overall cooling efficiency clearly depends on the defect/wall surface ratio and on the actual distribution of defects.

The extension to surfaces with many defects turns out to be quite natural. Clearly, such a scenario is not only closer to the characteristics

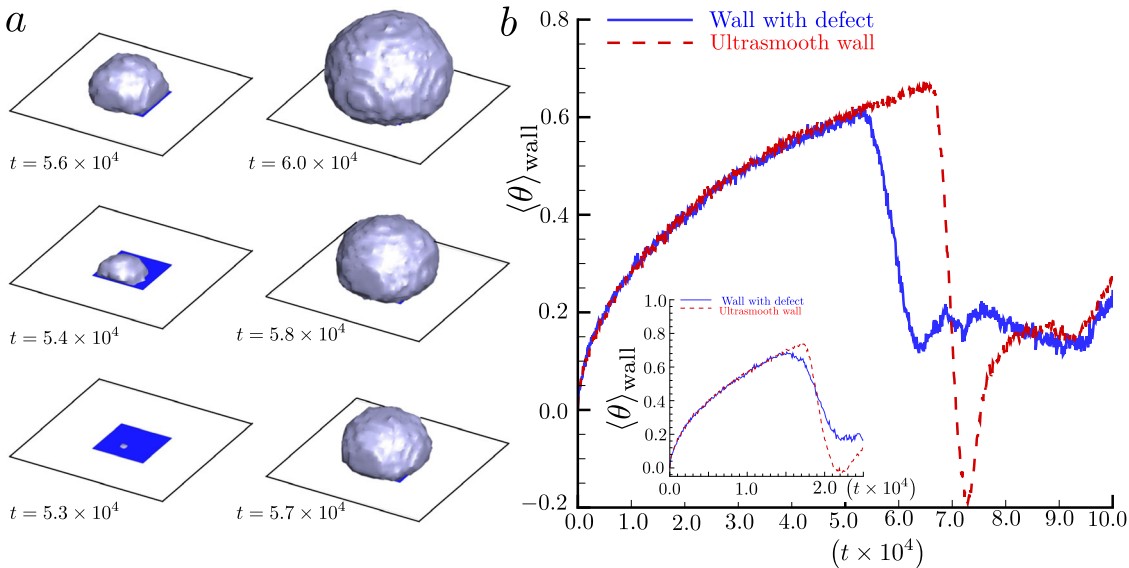

Fig. 3 | **Nucleation from a defect. a** Different snapshots of mesoscale simulations of the boiling process that originates from a vapor bubble nucleation on a small hydrophobic defect (blue squares) with $\phi = 105°$. The white complementary region has a hydrophilic wettability with $\phi = 30°$. The initial saturation temperature was $T_{\text{sat}} = 0.90$ and the externally imposed heat flux as $Q = 0.005$, the time instant is

reported on the left bottom side of the surface. **b** The mean reduced temperature as a function of time is reported both for a hydrophilic homogeneous wall (red line) and for the same surface augmented with a hydrophobic defect (blue line), the latter case refers to the simulation illustrated in the left panel. The inset reports the same quantities with a different externally imposed heat flux $Q = 0.01$.

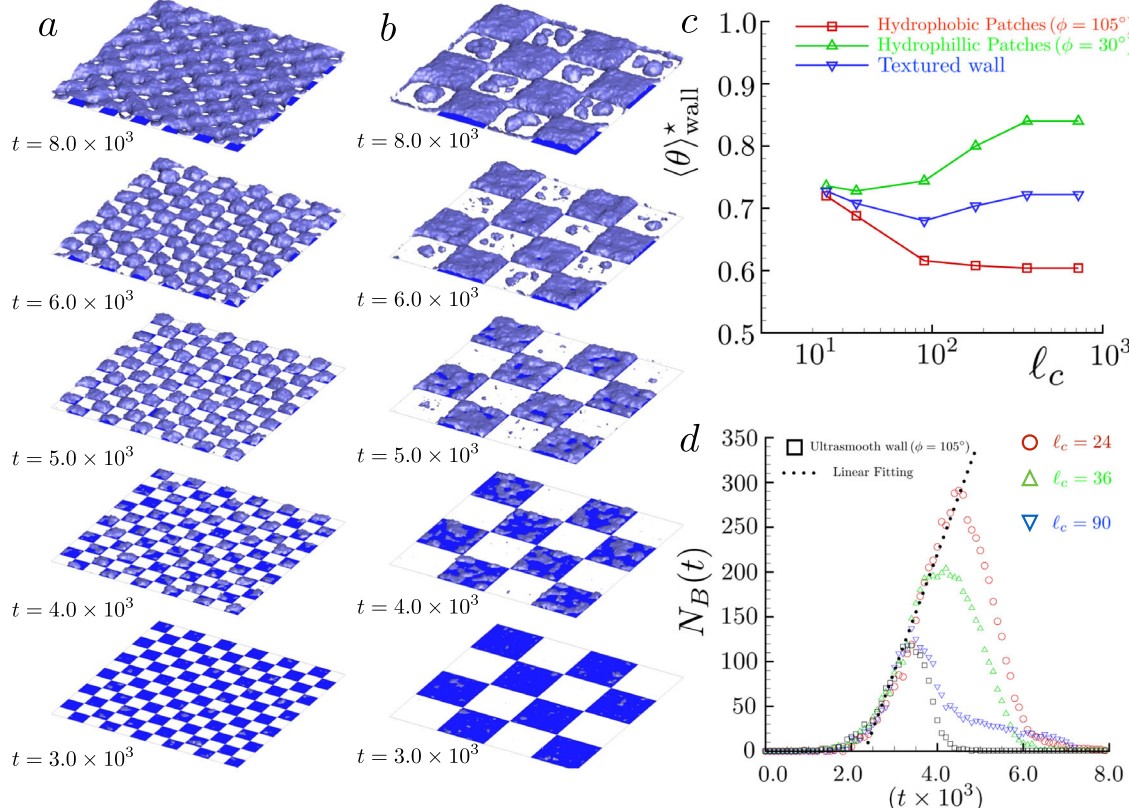

**Fig. 4 | Nucleation on structured hydrophobic/hydrophilic patches.**
**a, b** represent different snapshots in time of boiling simulations on structured hydrophobic/hydrophilic patches. The blue squares represent the hydrophobic region ($\phi = 105°$), while the white patches correspond to hydrophilic chemistry $\phi = 30°$, the time instant is reported on the left bottom side of the surface. In (**a**) the patch side is $\ell_c = 60$, while $\ell_c = 180$ in (**b**). **c** Reduced onset boiling temperatures vs the dimension of the squared hydrophobic (blue squares) and hydrophilic (white squares). The green line with upward-facing triangles refers to the mean reduced onset temperature evaluated on hydrophilic patches. The red line with red squares indicates the mean reduced onset temperature calculated on hydrophobic patches. The blue line with downward-facing triangles represents the reduced onset temperature of the whole structured wall. **d** The number of nucleated bubbles on a textured wall vs time. Different symbols refer to different values of $\ell_c$. Red circles for $\ell_c = 24$, green upward-facing triangles $\ell_c = 36$, blue downward-facing triangles $\ell_c = 90$. Black squares represent the hydrophobic homogeneous wall. The black dotted line is the linear fitting of the steady-state bubble production rate.

of common solid walls but also of great application interest[34,35]. In Fig. 4a, b the boiling process on a hydrophilic/hydrophobic chessboard is illustrated with different snapshots taken from numerical simulations. The simulation consists of a square wall with hydrophilic wettability $\phi = 30°$ (white patches) and hydrophobic spots (blue regions with $\phi = 105°$), here $Q = 0.01$. Different patch side lengths $\ell_c$ are investigated. The cubic simulation domain has a length $L = 720 = 0.53\,\mu m$ for all the considered values of $\ell_c$ except for the biggest one ($\ell_c = 720$) where a wall with side length $L = 1440 = 1.07\,\mu m$ has been considered in order to have patches as extended as the homogeneous cases considered in Fig. 1a, b. As observed in single defect simulations, the bubble nucleation starts from hydrophobic patches and develops with an expansion that is influenced by the pinning/depinning dynamics at the abrupt wettability contrasts. The coalescence dynamics are mainly driven by bubbles merging through the corners of hydrophobic patches. Such a process speeds up coalescence and promotes faster vapor nanofilm formation, especially for finer checkerboards.

In Fig. 4c the onset temperature is reported as a function of $\ell_c$. The red line with square symbols indicates $\langle\theta\rangle^\star_{wall}$ as evaluated on hydrophobic patches. $\langle\theta\rangle^\star_{wall}$ decreases with $\ell_c$ recovering the homogeneous wall value for large patches. Conversely, the onset temperature of the hydrophilic patches (green line with triangles) shows the opposite trend. Overall, the nucleation preferentially occurring on the hydrophobic zones can cool down the entire wall. For small values of $\ell_c$ ($\ell_c < 90 = 66.6\,nm$) the chessboard structures

homogenize onset temperatures. The blue line with downward-facing triangles represents the reduced onset temperature of the whole structured wall.

In macroscopic descriptions, one of the most relevant observable is the nucleation site density which determines the formation of the vapor film, the so-called boiling crisis[36]. It is directly linked to the density and geometry of chemical or geometric inhomogeneities that catalyze bubble formation. This concept is however elusive for ultrasmooth, chemically homogeneous walls, where the nucleation probability is uniform and vapor nuclei cannot exist in a stable form. In this context, the nucleation site density is better understood as an effective notion to account for the missing nanoscopic information on the phase transition. As a matter of fact, the experimentally observed nucleation sites are micrometrical objects which are well beyond the nucleation stage. The critical nuclei of pure liquids have a typical size in the range from a few to ten/twenty nanometers, depending on metastability degree. At those scales, pure vapor bubbles are strongly unstable, except for closed microcavities[22,37]. Actually, in open systems, only two thermodynamic stable conditions exist, namely the homogeneous liquid and vapor, respectively.

In order to reconcile the "macroscopic" view with the present stochastic description, Fig. 4d shows the time history of supercritical bubble numbers for different chessboards (see the caption for details). Apparently, the bubble number increases with the hydrophobic patches acting as nucleation sites. At the same time, as already discussed, the nucleation rate (slope of the $N_B$ vs time curve normalized by the

total hydrophobic area) is found to be constant and only determined by the hydrophobic regions' wettability. The larger number of bubbles for the cases with more defects is directly related to the pinning on defect boundaries, which prevents/slows down the coalescence, retarding the formation of the vapor nanofilm. As for ultrasmooth walls, the increasing wall bubble density triggers repeated coalescence/collapse events that lead to a fast decrease of their number, as also observed in the context of cavitation[15,38].

## Discussion

The present analysis sheds new light on the longstanding issue of experimental onset temperatures way smaller than predicted by theory. In order to reconcile theory and experiments, the presence of unobserved microcavities on the solid wall is generally postulated, based on the practical impossibility to control the purity of the samples (both the liquid and the solid surface) down to contaminants as small as a few nanometers in size. Actually, we show that, by applying the classical approach to our simulation data of boiling over ideally flat, chemically homogenous surfaces, the same overestimate of the onset temperature is found. The mismatch has a twofold origin: on one hand, the classical approach is based on homogenous nucleation theory (i.e. in the absence of confining walls). On the other hand, the bubble nuclei are small enough to suspect that the surface tension could depend on the bubble radius, as originally suggested by Tolman. In fact, taking into account only the presence of the wall is found insufficient to interpret the simulations, which seem instead consistent with the existence of Tolman lengths on the order of fractions of nanometers.

Despite the Tolman corrected CNT providing a consistent interpretation of the data, it still needs a substantial experimental input, namely the onset temperature and the Tolman length. It should be emphasized that the energy contributions that could affect the results of the simulations through a modification of the nucleation barrier and the intensities of the thermal fluctuations are multiple. In particular, the modification of the Tolman length in finite temperature contexts, the structure of the density field at the wall, and possible line stresses associated with the triple line. This highlights the importance of a reliable simulative approach framed by a sound physical model. Based on an effective Fluctuating Diffuse Interface approach, we demonstrated the multiscale nature of boiling in action. Besides the predictive capabilities, the proposed methodology offers unprecedented computational efficiency, enabling it to address the poorly explored nano-to-micro range of scales. The approach is shown able to bridge the spatial/temporal gap between hydrodynamical and atomistic scales allowing the development of a physically based multiscale methodology in the next future.

What stands out is that, at variance with current understanding, sparse nanoscale defects (of favorable, hydrophobic wettability) are sufficient to trigger the nucleation at onset temperatures far below those of perfectly uniform, hydrophilic surfaces. Then, the fast evaporation expands the bubble to experimentally detectable sizes, thus explaining the puzzling experimental observation of an unexpectedly low onset temperature.

We stress that, from a general perspective, the ideally smooth wall we have considered can be seen as an extreme case to investigate the issue of the low onset temperatures that could be further pursued considering nucleation on atomically smooth, liquid–liquid surfaces[39]. Attention can be also called to the role of dissolved gas in the nucleation process which is known to anticipate nucleation[40]. Recent contributions are available in the literature, where dissolved gas is considered in the context of the diffuse interface, deterministic models[41–43]. Similar approaches can be easily included in the present stochastic description and are considered in currently ongoing research.

As a final comment, we would like to stress that numerical simulations with the typical heat fluxes used in pool boiling experiments (typical critical heat flux of water $Q \sim 1\,\mathrm{MWm^{-2}}$) can be in principle be realized. It should be noted, however, that as the temperature rises at constant pressure, the fluid becomes progressively more metastable with an increasingly reduced nucleation barrier. As a consequence, the nucleation time, exponential in the barrier height, goes from extremely large near saturation to vanishingly small as spinodal conditions are approached. The heat flux determines the rate at which the temperature rises. If the temperature rises slowly, even the large barrier at small temperatures can be crossed, since sufficient time is available. If the temperature rises fast, on the opposite, the temperature keeps rising before the barrier can be crossed, up to a point where the barrier becomes small enough such that nucleation takes place before the temperature can grow further. Hence, due to the exponential dependency on the barrier, the onset boiling time decreases at, approximately, an exponential rate by increasing the heat flux. This cartoon is consistent with the data that show an increase in the boiling onset time from nanoseconds to microseconds with a heat flux reduction from $Q = 75.6\,\mathrm{MWm^{-2}}$ to $Q = 7.56\,\mathrm{MWm^{-2}}$. Therefore, reducing the heat flux to $Q \sim 1\,\mathrm{MWm^{-2}}$ though conceptually feasible, is, extremely arduous computationally. In perspective, rare event techniques can be the proper tool to approach such kinds of problems.

## Methods

### Diffuse interface and fluctuating hydrodynamics

The adopted mesoscale model is based on the Van der Waals energy functional accounting for solid-fluid interaction energy. In this approach, the local description of a simple fluid is augmented by including in the free energy a capillary contribution that is proportional to the density gradient squared,

$$F[\rho,T] = \int_V f_b(\rho,T) + \frac{\lambda}{2}|\boldsymbol{\nabla}\rho|^2 \, \mathrm{d}V + \oint_{\partial V} f_w(\rho,T)\,\mathrm{d}V \qquad (2)$$

where $f_b(\rho,T)$ is the bulk free energy density, depending on mass density $\rho(\mathbf{x})$ and temperature $T(\mathbf{x})$. $\lambda$ is the capillary coefficient controlling both the liquid/vapor interface thickness and the surface tension[44]. The diffuse interface description naturally includes the curvature-dependent (Tolman) correction to the surface tension, and reproduces the correct scaling for capillary waves and fluctuation spectra[29,45,46]. The solid/fluid free energy density $f_w(\rho,T)$ is defined as[15]

$$f_w[\rho,T] = f_w[\rho_V] - \cos\phi \int_{\rho_V}^{\rho} \sqrt{2\lambda\left[\omega_b(\rho^\star,T) - \omega_b(\rho_V)\right]}\,\mathrm{d}\rho^\star, \qquad (3)$$

where $\rho_V$ is the saturation density of the vapor and $\omega_b = f_b - \mu_{eq}\rho$ is the Landau Grand Potential density with $\mu_{eq}$ the equilibrium chemical potential.

Equation (2) describes the equilibrium properties of the inhomogeneous fluid. Concerning dynamics, hydrodynamical equations need to be augmented with stochastic fluxes to account for thermal fluctuations in a continuum description. The ensuing balance for mass, momentum, and energy read

$$\frac{\partial \rho}{\partial t} + \boldsymbol{\nabla} \cdot (\rho\mathbf{u}) = 0\,,$$
$$\frac{\partial \rho\mathbf{u}}{\partial t} + \boldsymbol{\nabla} \cdot (\rho\mathbf{u} \otimes \mathbf{u}) = \boldsymbol{\nabla} \cdot \boldsymbol{\Sigma} + \boldsymbol{\nabla} \cdot \delta\boldsymbol{\Sigma}\,, \qquad (4)$$
$$\frac{\partial E}{\partial t} + \boldsymbol{\nabla} \cdot (\mathbf{u}E) = \boldsymbol{\nabla} \cdot (\mathbf{u} \cdot \boldsymbol{\Sigma} - \mathbf{q}) + \boldsymbol{\nabla} \cdot (\mathbf{u} \cdot \delta\boldsymbol{\Sigma} - \delta\mathbf{q})\,.$$

The deterministic fluxes are determined by non-equilibrium thermodynamic arguments leading to[44]

$$\boldsymbol{\Sigma} = \left[ -p + \frac{\lambda}{2}|\boldsymbol{\nabla}\rho|^2 + \rho\boldsymbol{\nabla}\cdot(\lambda\boldsymbol{\nabla}\rho) \right]\mathbf{I} - \lambda\boldsymbol{\nabla}\rho \otimes \boldsymbol{\nabla}\rho +$$
$$+ \eta_1(\boldsymbol{\nabla}\mathbf{u} + \boldsymbol{\nabla}\mathbf{u}^T) + \eta_2\boldsymbol{\nabla}\cdot\mathbf{u}\mathbf{I},$$
$$\mathbf{q} = \lambda\rho\boldsymbol{\nabla}\rho\boldsymbol{\nabla}\cdot\mathbf{u} - k\boldsymbol{\nabla}T, \tag{5}$$

where the transport coefficients $\eta_1, \eta_2, k$, represent the fluid's first and second viscosity coefficients and thermal conductivity, respectively. The fluctuation-dissipation balance provides the stochastic fluxes[15,29]

$$\langle\delta\mathbf{q}(\hat{x},\hat{t})\rangle = 0 \quad \langle\delta\boldsymbol{\Sigma}(\hat{x},\hat{t})\rangle = 0,$$
$$\langle\delta\boldsymbol{\Sigma}(\hat{x},\hat{t}) \otimes \delta\boldsymbol{\Sigma}^\dagger(\tilde{x},\tilde{t})\rangle = \mathbf{Q}^{\boldsymbol{\Sigma}}\delta(\hat{x}-\tilde{x})\delta(\hat{t}-\tilde{t}),$$
$$\langle\delta\mathbf{q}(\hat{x},\hat{t}) \otimes \delta\mathbf{q}^\dagger(\tilde{x},\tilde{t})\rangle = \mathbf{Q}^{\mathbf{q}}\delta(\hat{x}-\tilde{x})\delta(\hat{t}-\tilde{t}), \tag{6}$$

where

$$\mathbf{Q}^{\boldsymbol{\Sigma}}_{\alpha\beta\nu\eta} = 2k_BT\left[\eta_1(\delta_{\alpha\nu}\delta_{\beta\eta} + \delta_{\alpha\eta}\delta_{\beta\nu}) + \eta_2\delta_{\alpha\beta}\delta_{\nu\eta}\right],$$
$$\mathbf{Q}^{\mathbf{q}}_{\alpha\beta} = 2k_BT^2k\delta_{\alpha\beta}, \tag{7}$$

with $k_B$ the Boltzmann constant.

System (6) is closed using the van der Waals equation of state (EoS) for the pressure $p(\rho,T)$ and the internal energy density $u(\rho,T) = u_b(\rho,T) + \lambda/2|\boldsymbol{\nabla}\rho|^2$, with $u_b$ the bulk internal energy density.

Keeping the same symbols, the thermodynamic quantities $p$ and $u_b$ can be expressed in reduced variables, i.e., normalizing pressure, temperature, and density with the corresponding critical values, $p_c, \theta_c, \rho_c$, and the internal energy density with the critical pressure, leading to[47]

$$p = \frac{8\rho}{3-\rho}\theta - 3\rho^2, u_b = 8\rho\theta - 3\rho^2. \tag{8}$$

## Numerical method

The set of Eqs. (4) is represented by a system of stochastic partial differential equations. In this work, they have been discretized on a uniformly spaced staggered grid. Scalar fields, namely density, and temperature are located at the cell center. The components of vector fields are located at the center of the perpendicular face. We performed the time integration by adopting a second-order Runge-Kutta explicit scheme, see Supplementary Informations for further details. As far as the deterministic part is concerned, there are three important numerical aspects that should be taken into account. (i) Acoustic waves induced by fluid compressibility. (ii) Capillarity (described by the contribution of density gradients in the stress tensor). (iii) Diffusion induced by viscous forces and thermal conduction. For an explicit time integrator, there are timestep limitations that must be ensured in order to preserve the stability of the scheme. The stability bounds of the scheme have been identified for a Runge-Kutta temporal integrator in ref. 48, where a stability analysis of the linearized dynamics was performed. We used a time step coherently with the aforementioned analysis. Concerning stochastic dynamics, particular attention should be paid to both the temporal integrator and the spatial discretizations[49]. The numerical scheme should be able to reproduce the fields' statistical properties at the discrete level. In other words, the fluctuation-dissipation balance must be preserved in the adopted spatial discretization. As discussed in the paper[49] this is the case of central staggered discretization and second order Runge-Kutta integrator, as we employed in this work. The

convergence of the adopted scheme has been discussed in our previous works[15,29].

## Reference quantities

The quantities reported in the text and in the figures are dimensionless, with the reference value chosen as the reduced units in Van der Waals EoS, see also SI for further details. By considering water, $T_c = 647$ K for temperature, $p_c = 22$ MPa for pressure, $\rho_c = 196.8$ Kgm$^{-3}$ for density. Consistently, the reference length is $L_R = (k_BT_c/p_c)^{1/3} = 0.74$ nm, the reference velocity $= u_R = (p_c/\rho_c)^{1/2} = 334.8$ m/s the reference time $t_R = L_R/u_r = 2.21$ ps and the reference heat flux $q_R = p_cu_r = 7.56$ GWm$^{-2}$. The value of the capillary coefficient is fixed to $\lambda = 5.3 \times 10^{-16}$ m$^7$s$^{-1}$kg$^{-1}$ to reproduce the correct value of the surface tension of water ($\sigma = 0.072$ N/m). This value of the capillary coefficient enforces a liquid/vapor interface thickness at ambient conditions of $\epsilon = 1.3$ nm in accordance with experimental observation[50]. The transport coefficients as viscosity $\eta_1, \eta_2(\rho,T)$ and thermal conductivity $k(\rho,T)$ are taken from the International Association for the Properties of Water and Steam (IAPWS)[51].

## Tolman correction to classical nucleation theory

In liquid/vapor thermodynamic equilibrium, at fixed temperature $T$ the two phases share the same chemical potential $\mu$. In this condition the Gibbs-Duhem equality expresses the chemical potential in terms of the pressure $p$

$$\mu(p,T) - \mu_{\text{sat}} = \int_{p_{\text{sat}(T)}}^{p}\frac{d\hat{p}}{\rho(\hat{P},T)}. \tag{9}$$

The above equation can be specialized for the vapor phase (described as an ideal gas with specific constant $\mathcal{R}$)

$$\mu_v = \mu_{\text{sat}} + \mathcal{R}T\ln\frac{p_v}{p_{\text{sat}}(T)}, \tag{10}$$

and for the liquid (incompressible) state

$$\mu_l = \mu_{\text{sat}} + \frac{p_l - p_{\text{sat}}(T)}{\rho_l}, \tag{11}$$

respectively. By enforcing the chemical equilibrium $\mu_v = \mu_l = \mu$ at the given temperature $T$ the following relationship between the vapor ($p_v$) and liquid pressure ($p_l$) is established

$$p_v = p_{\text{sat}}(T)\exp\left(\frac{p_l - p_{\text{sat}}(T)}{\mathcal{R}T\rho_l}\right). \tag{12}$$

Let us now focus on the free energy (Grand Potential) difference $\Delta\Omega$ associated with the formation of a vapor spherical cup of radius $R$ on a flat solid surface with contact angle $\phi$,

$$\Delta\Omega = \left(-\frac{4}{3}\pi\Delta P(\mu,T)R^3 + 4\pi\sigma(R,\mu,T)R^2\right)\psi(\phi), \tag{13}$$

with $\Delta P = p_v - p_l$, $\sigma$ the liquid/vapor surface tension, and $\psi(\phi) = 1/4(1 + \cos\phi)^2(2 - \cos\phi)$ a geometric factor accounting for the contact angle[15]. In the above equation, the surface tension is taken to depend on the bubble radius, as required[9–11,22] in order to correctly estimate the nucleation energy barrier,

$$\sigma(R) = \sigma_0\left(1 - \frac{2\delta}{R}\right), \tag{14}$$

where $\sigma_0$ is the liquid/vapor surface tension of the planar interface and $\delta$ the Tolman length[52].

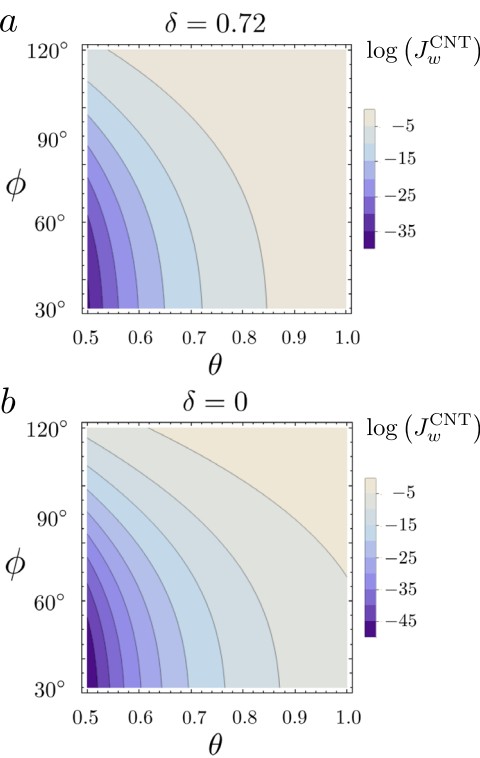

**Fig. 5 | CNT Nucleation Rate.** Heterogeneous nucleation rate as a function of the reduced temperature $\theta$ and the contact angle $\phi$. **a** CNT corrected with the Tolman length $\delta = 0.72$. **b** Simple CNT with $\delta = 0$.

The energy barrier $\Delta\Omega^\dagger$ for the liquid/vapor phase transitions is attained at the critical radius $R_c$, i.e., $d\Delta\Omega/dR(R_c) = 0$, implying

$$\Delta P^\dagger = \frac{2\sigma}{R_c} + \frac{d\sigma}{dR}\Big|_{R_c} = \frac{2\sigma_0}{R_c}\left(1 - \frac{\delta}{R_c}\right) \tag{15}$$

and

$$R_c = \frac{\sigma_0 + \sqrt{\sigma_0^2 - 2\Delta P\sigma_0\delta}}{\Delta P}. \tag{16}$$

Eventually, the Tolman corrected energy barrier reads

$$\Delta\Omega^\dagger(T,\phi,\delta) = \frac{4}{3}\pi R_c^2\sigma_0\left(1 - \frac{4\delta}{R_c}\right)\psi(\phi). \tag{17}$$

At constant liquid pressure, as in boiling conditions, the energy barrier $\Delta\Omega^\dagger$ is a function of temperature and contact angle and depends on the Tolman length. In particular, the barrier is a decreasing function of temperature, since $p_{sat}$ is an increasing function of $T$ and, approaching saturation conditions ($p_l \to p_{sat}$), $\Delta\Omega^\dagger \to \infty$. Given the energy barrier, the nucleation rate can be expressed as[15,53]

$$J_w^{CNT}(T,\phi,\delta) = n_l^{2/3}\frac{1 + \cos\phi}{2}\sqrt{\frac{2\sigma}{\pi m}}\exp\left(\frac{-\Delta G^\dagger}{k_B T}\right), \tag{18}$$

where $n_l$ and $m$ are the number density and mass of liquid molecules, respectively. The nucleation rate is reported in Fig. 5 as a function of the contact angle $\phi$ and the reduced temperature $\theta = (T - T_{sat})/(T_{spin} - T_{sat})$ with $T_{sat} = 0.95$ and $T_{spin} = 0.9625$. Figure 5a depicts a case with Tolman length $\delta = 0.74 \simeq 0.5$ nm, in comparison with the standard theory ($\delta = 0$), reported in Fig. 5b. Apparently, the nucleation rate is extremely sensitive to changes in $\delta$.

## Tolman correction of the onset temperature

The classical literature on boiling[3] often makes reference to a so-called onset temperature, defined as the temperature for which the classical nucleation theory predicts the same onset nucleation rate, observed/estimated from experiments, $J_w^\star(T_{onset}) = J_w^\star(exp)$, where $J_w^\star(T)$ is the homogenous nucleation rate.

When we apply the same approach, correcting the theoretical expression of the rate to account for heterogeneous nucleation and using the experimental nucleation rate from the simulations, it turns out that the onset temperature we obtain is significantly over-estimated (Fig. 1d). Hence we decided to include the Tolman length in the description, by invoking Eq. (18) to estimate $\delta$ using onset temperature and nucleation rate as inputs taken from the numerical data. Overall the results we found provide reasonable values for the estimated Tolman length, see Supplementary Table 1. The conclusion is that, conversely, in order to estimate the correct onset temperature from the estimated nucleation rate one should provide an appropriate value for such an elusive quantity as the Tolman length, which we show to be crucial to an accurate description of the nucleation process.

## Data availability

The raw data for the plots generated in this study have been deposited in the figshare database under the accession link https://doi.org/10.6084/m9.figshare.24171582. Additional data that support the findings of this study are available from the corresponding author upon request.

## Code availability

The numerical codes used in this study have been deposited in the Zenodo database under the accession link https://doi.org/10.5281/zenodo.8365891. Additional codes are available from the corresponding authors upon request.

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

## Acknowledgements

C.M.C. has been partially supported by the Sapienza 2022 Funding Scheme, Project No. RG1221815884CB65. C.M.C. and M.G. have received financial support from ICSC – Centro Nazionale di Ricerca in "High Performance Computing, Big Data and Quantum Computing", funded by European Union – NextGenerationEU. M.M., A.G., M.G., and J.dC. have been supported by the UK LEVERHULME Fund grant AMuSE RPG-2021-262. Support is acknowledged from DECI 17 SOLID project for resource Navigator based in Portugal at https://www.uc.pt/lca/ from the PRACE aisbl (PI M.G.); CINECA award under the ISCRA initiative, for the availability of high-performance computing resources and support ISCRA-B FHDAS (PI M.G.).

## Author contributions

M.G., F.M., A.G., M.M., J.dC., and C.M.C. designed the research. M.G., F.M., and C.M.C. developed the model; M.G. and F.M. wrote the codes; M.G. ran the simulations and wrote the paper draft; M.G., F.M., A.G.,

M.M., J.dC., and C.M.C. performed research, interpreted the results, and revised the paper.

## Competing interests

The authors declare no competing interests.
