## [Peer Review File · Nature Communications]

REVIEWER COMMENTS

Reviewer #1 (Remarks to the Author):

This is a very nice paper suggesting a possible explanation for the low superheat levels that are required in practice to generate vapor bubbles compared to theoretical estimates. The authors use the fluctuating diffuse interface approach to study the problem. Overall, I am very enthusiastic about this paper, and I think it is well deserving of publication in Nature Communications. To improve what is already an excellent manuscript, I would encourage the authors to address the following minor comments in a revision.

1) One possible explanation that emerges from the authors' approach is based on random hydrophobic spots on the surface. Do the authors know if this occurs in practice and what would explain these spots? In my mind, these spots are more difficult to imagine than roughness perturbations.

2) For reproducibility purposes, the paper would benefit from some details about the numerical method used to carry out the simulations.

3) One aspect that is also likely to be important in the nucleation process is the presence of non-condensable gases as shown in doi:10.1063/5.0131165 in the context of cavitation. A discussion on this point would be helpful.

Reviewer #2 (Remarks to the Author):

The paper titled "A nanoscale view of the origin of boiling and its dynamics" by Mirko Gallo et al. presents a theoretical study on bubble nucleation phenomena.

The authors show that a continuum model based on fluctuating hydrodynamics and a diffuse interface approach can predict heterogeneous spontaneous nucleation temperature close to those observed experimentally on boiling surfaces.

This work addresses an important open question in thermal science: what are the phenomena triggering the nucleation of a bubble on a heated surface?

As mentioned by the authors, it is currently impossible to visualize or measure these phenomena with the spatial and temporal resolution necessary to capture all the space and time scales of the physical process. The highest resolutions that advanced techniques can achieve are in the order of a few microns and a few hundred microseconds.

The authors attempt to theoretically reconstruct the phase of a bubble life cycle that goes from sub-nanometric length scales to micrometric scales (starting from which, continuum fluid mechanics can explain the phenomena observed experimentally). Note that the authors are not to blame if there are no experimental data to validate their model. However, they could provide some dimensional results (see MAJOR COMMENT #2).

The believe that this work present an important contribution to the field, and deserves to be published once the authors have properly addressed the comments and questions below.

MAJOR COMMENTS

MAJOR COMMENT #1

According to classical theories, bubble nucleation at low temperatures (much lower than those predicted by heterogenous spontaneous nucleation models) is possible due to the presence of micrometric cavities on a boiling surface, where vapor or, to start, non-condensable gases gets trapped. The smaller is a cavity, the higher is the temperature required for a vapor embryo to start to grow. In absence of such micrometric imperfections (i.e., on a perfectly nanosmooth surface), the nucleation temperature would increase. However, the authors point out that even on “nominally” nano-smooth surfaces, scientists have measured very low nucleation temperatures, comparable to those measured on surfaces with micrometric roughness (e.g., commercial steel or copper). This argument is supported by Refs. [5,6]. These observations motivate the need for a heterogenous spontaneous nucleation model.

In the reviewer experience, it is unlikely that the surfaces used in Refs. 5 or 6 are completely free of micron-scale cavities. In particular, when using glass substrates, such as is Ref. 5, residual microcavities may be present on the glass surface even after the polishing process. Such microcavities are partially but not entirely covered by the thin-film coating, which is ~1 micron thick (e.g., see Richenderfer, A., Kossolapov, A., Seong, J.H., Saccone, G., Demarly, E., Kommajosyula, R., Baglietto, E., Buongiorno, J. and Bucci, M., 2018. Investigation of subcooled flow boiling and CHF using high-resolution diagnostics. *Experimental Thermal and Fluid Science*, 99, pp.35-58).

In summary, while the reviewer does not question the significance of the work, it is not convinced that the motivation of the work is justified properly.

MAJOR COMMENT #2

The results are presented in non-dimensional unit only. The authors discuss how to get back to dimensional units in the methods section. However, the reviewer thinks it would be appropriate to present some results in dimensional form to help the reader appreciate the significance of the work.

Related to this comment, is also the observation that the heat fluxes used by the authors seem to be quite high. For instance, $Q = 0.001$ (which is the lowest non-dimensional heat flux) “corresponds to a few MW/m², the typical heat flux used in experiments.” In most experiments, onset of nucleate boiling occurs at fraction of a MW/m². In fact, the critical heat flux of water in pool boiling conditions is in the order of 1 MW/m². Can the authors provide results for Q values corresponding to ~ 0.1 MW/m² (i.e., onset of boiling conditions)?

MAJOR COMMENT #3

One may argue that when boiling has started, residues of vapor trapped on cavities or on the surface may serve as nucleation sites. In fact, experiments have shown that this nucleation sites can be identified (e.g., using infrared or other optical techniques) and are quite stable. Thus, the theory developed by the authors may apply to the onset of boiling, when there is potentially no micron-scale vapor (or non-condensable gas) structure.

MINOR COMMENTS:

1. The abstract is very lengthy. I believe that Nature Communications sets a word limit on the abstract. I suggest the authors to simplify the abstract by focusing on the novelty and the impact of their work. The introductory part of the abstract discussing the motivation of the work can be shortened significantly, as it is detailed discussed in the introduction.

2. The introduction is also quite lengthy. I advise the authors to write more concisely.

3. The first section in the Results (i.e., how surface wettability influences boiling) can be structured in clearer way. For instance: It is not clear why the inset in Figure 2 is not presented with Figure 1; the definition of J_w is provided after its values are discussed in the text.

4. There is an oversight in the intro that should be fixed, with respect to Ref. 18. The temporal resolution in ref 18 is ~ 1 millisecond, not microsecond. Infrared cameras are much slower than high-speed video camera. However, in Ref. 18, the authors measured the temperature at the back side of a silicon wafer (i.e., the side opposite to the one where boiling occurs). The authors could find better examples of high-speed video or high-speed infrared thermometry that focused on bubble nucleation.

5. How do you explain the fact that, in Figure 3, the temperature on the ultra-smooth wall goes below 0, i.e., below the saturation temperature?

Replay Referee # 1

NCOMMS-23-29819-T.

A nanoscale view of the origin of boiling and its dynamics. By Gallo et al.

This is a very nice paper suggesting a possible explanation for the low superheat levels that are required in practice to generate vapor bubbles compared to theoretical estimates. The authors use the fluctuating diffuse interface approach to study the problem. Overall, I am very enthusiastic about this paper, and I think it is well deserving of publication in Nature Communications. To improve what is already an excellent manuscript, I would encourage the authors to address the following minor comments in a revision.

We thank the referee for the time spent reviewing our paper and for the kind words of appreciation for the work. The referee provides some interesting comments which give us the opportunity to better explain some aspects of our work and to improve its exposition and clarity.

All required changes have been made to the new version of the manuscript and highlighted in blue for the referee's convenience.

We provide below point-by-point answers to all the questions raised in the review.

Points raised by the Referee

- 1) One possible explanation that emerges from the authors' approach is based on random hydrophobic spots on the surface. Do the authors know if this occurs in practice and what would explain these spots? In my mind, these spots are more difficult to imagine than roughness perturbations.***

The possibility for a solid surface to have random spots of different wettability properties may arise in different scenarios. In the experimental work of ref [5], the authors observed that aged and fresh surfaces behave differently in terms of the number of nucleated bubbles, with up to an order magnitude more nucleated bubbles when considering aged surfaces. This suggests the possibility of changing the chemical surface properties due to oxidation processes, that at the microscopic level reflect in small nanometric regions of different wettability. Notably, Frenkel in his early works in 1940 already proposed the idea that inhomogeneities can be responsible for heterogeneous nucleation (J Frenkel. Kinetic theory of liquids. Dover, 1955.), introducing the concept of Frenkel's island, poorly wettable microscopic patches due to impurities and/or surfactants. SEM imaging of aged surfaces, see Fig. 8 of ref [5], revealed the density of the so-called Frenkel's islands (up to 10^8 islands per cm^2) with a characteristic size of about $300 \times 300 \text{ nm}^2$.

Similarly, the contamination of the liquid by parts of amphiphilic molecules can lead to different wettability spots. This is the case of lipids which, if present in water, tend to arrange themselves in micellar structures. For energetic reasons, the heads of the lipids are always facing the aqueous environment (strongly hydrophilic), and therefore at certain

levels of concentrations, they are arranged as closed structures with the heads facing the water and the tails intertwined with each other. Therefore, where present, they act locally as a highly hydrophilic region. The presence of other hydrophobic chemical compounds that contaminate the liquid, such as Perfluorocarbons, can conversely induce hydrophobic regions.

We thank the Referee for this important observation which gives us the possibility to clarify an important point of our work. In this regard, we take up a comment (reported below) also made to Referee # 2, in which we elucidate more clearly the motivation of our work.

We believe, however, that our modeling suggests that at the nucleation level what really matters is the local free energy barrier that a heterogeneous surface possesses. From this point of view, the checkerboard configurations analyzed with alternating hydrophobic and hydrophilic chemistries clearly show that the nucleation rate is determined only by the hydrophobic zones having lower activation energy. Such zones with low activation energy could be seen as nucleation sites in a coarse-grained view of the liquid/vapor phase transition.

We can then speculate that a similar mechanism occurs on walls having certain roughness and/or dissolved gas nuclei.

In the former case, the situation is quite straightforward, because CNT (Blander, Milton, and Joseph L. Katz. "Bubble nucleation in liquids." *AICHE Journal* 1975) shows a decrease in barriers when geometric imperfections are present.

In the second, it is easy to verify that the presence of gas decreases the nucleation barrier (opening a bottle of sparkling water is a nice experimental validation). As a matter of fact, the dissolved gas content modifies the position of the spinodal line with respect to the pure liquid (Wang, Yuliang, et al. "Giant and explosive plasmonic bubbles by delayed nucleation." *Proceedings of the National Academy of Sciences* 2018), making the nucleation barrier lower.

From this point of view, the smooth wall can be seen as an extreme case study to investigate the phenomenon of low onset temperatures.

2) For reproducibility purposes, the paper would benefit from some details about the numerical method used to carry out the simulations.

In the new version of the manuscript, we have added a dedicated section in Materials and Methods where the main numerical aspects are treated. Furthermore, more references are added for both the treatment of the deterministic part of the equations and the stochastic one. For that stochastic dynamics we have also provided some additional details in the Supplementary Informations.

3) One aspect that is also likely to be important in the nucleation process is the presence of non-condensable gases as shown in doi:10.1063/5.0131165 in the context of cavitation. A discussion on this point would be helpful.

The Referee is right. The nucleation of vapor bubbles in metastable liquids is strongly influenced by the presence of gas. Persistent gas nuclei on the one hand can act as nucleation sites, and on the other hand change the spinodal limits of the liquid, which tend increasingly toward the saturation line as the concentration of dissolved gas in the liquid increases (Wang, Yuliang, et al. "Giant and explosive plasmonic bubbles by delayed

nucleation." Proceedings of the National Academy of Sciences 2018). The scenario is even more complex when considering the contextual presence of a solid surface.

The mechanism of bubble nucleation in a multicomponent fluid can in principle be tackled with modeling similar to the one adopted in the present work. In the same diffuse interface framework the addition of a second specie mimicking the dissolved gas can be introduced. Recently, deterministic diffuse interface models have been proposed in this direction. Along this line, it is worth mentioning the recent work aiming to estimate the tensile limit of water in the presence of dissolved gases (Mukherjee, Saikat, and Hector Gomez. "Effect of dissolved gas on the tensile strength of water." *Physics of Fluids* 2022).

Therefore it is reasonable to think of extending them to the context of fluctuating hydrodynamics to evaluate the effects of dissolved gases in the boiling process, a stimulating topic for future works. However, the main motivation of this work lies in showing through a non-equilibrium nucleation model that even in the limiting and ideal case of ultra-smooth walls alone, lower onset temperatures are observed than classical predictions, and that these temperatures are closely related to the chemistry of the wall and the intensity of external heat fluxes. Clearly, the presence of dissolved gas would result in a further decrease in onset temperatures for nucleation. Thanking the Referee for the interesting comment, we have added a dedicated discussion of this point in the new version of the manuscript by including the work mentioned.

In conclusion, we again thank the Referee for the work spent reviewing our paper. We believe that the suggestions provided have increased the quality of the paper in terms of both content, exposition, and clarity. Based on the Referee's comments, we are now confident that the paper can be ready for publication in Nature Communications.

Replay Referee # 2

NCOMMS-23-29819-T.

A nanoscale view of the origin of boiling and its dynamics. By Gallo et al.

The paper titled “A nanoscale view of the origin of boiling and its dynamics” by Mirko Gallo et al. presents a theoretical study on bubble nucleation phenomena.

The authors show that a continuum model based on fluctuating hydrodynamics and a diffuse interface approach can predict heterogenous spontaneous nucleation temperature close to those observed experimentally on boiling surfaces.

This work addresses an important open question in thermal science: what are the phenomena triggering the nucleation of a bubble on a heated surface?

As mentioned by the authors, it is currently impossible to visualize or measure these phenomena with the spatial and temporal resolution necessary to capture all the space and time scales of the physical process. The highest resolutions that advanced techniques can achieve are in the order of a few microns and a few hundred microseconds.

The authors attempt to theoretically reconstruct the phase of a bubble life cycle that goes from sub-nanometric length scales to micrometric scales (starting from which, continuum fluid mechanics can explain the phenomena observed experimentally).

Note that the authors are not to blame if there are no experimental data to validate their model. However, they could provide some dimensional results (see MAJOR COMMENT #2).

The believe that this work present an important contribution to the field, and deserves to be published once the authors have properly addressed the comments and questions below.

We thank the referee for their work on our paper and for acknowledging its relevance for the boiling community. Extremely interesting ideas emerge from the Referee’s letter which gives us the opportunity to clarify some aspects of our work and to improve its exposition and clarity at the same time. All required changes have been made to the new version of the manuscript and are highlighted in blue for the Referee’s convenience.

What follows are point-by-point answers to all the questions raised in the report.

Points raised by the Referee

MAJOR COMMENTS

MAJOR COMMENT #1

According to classical theories, bubble nucleation at low temperatures (much lower than those predicted by heterogenous spontaneous nucleation models) is possible due to the presence of micrometric cavities on a boiling surface, where vapor or, to start, non-condensable gases gets trapped. The smaller is a cavity, the higher is the temperature required for a vapor embryo to start to grow. In absence of such micrometric imperfections (i.e., on a perfectly nanosmooth surface), the nucleation temperature would increase. However, the authors point out that even on

“nominally” nano-smooth surfaces, scientists have measured very low nucleation temperatures, comparable to those measured on surfaces with micrometric roughness (e.g., commercial steel or copper). This argument is supported by Refs. [5,6]. These observations motivate the need for a heterogenous spontaneous nucleation model.

In the reviewer experience, it is unlikely that the surfaces used in Refs. 5 or 6 are completely free of micron-scale cavities. In particular, when using glass substrates, such as is Ref. 5, residual microcavities may be present on the glass surface even after the polishing process. Such microcavities are partially but not entirely covered by the thin-film coating, which is ~1 micron thick (e.g., see Richenderfer, A., Kossolapov, A., Seong, J.H., Saccone, G., Demarly, E., Kommajosyula, R., Baglietto, E., Buongiorno, J. and Bucci, M., 2018. Investigation of subcooled flow boiling and CHF using high-resolution diagnostics. Experimental Thermal and Fluid Science, 99, pp.35-58).

In summary, while the reviewer does not question the significance of the work, it is not convinced that the motivation of the work is justified properly.

We sincerely thank the Referee for bringing us to the attention of the cited paper (now added to the references list) and for giving us the opportunity to explain more clearly the motivation of our work.

Our primary objective is to show through a non-equilibrium nucleation model that also on ideally ultra-smooth walls the onset boiling temperature may remain below that expected from classical theories. Along this line, it follows that the mere mechanism of heterogeneous nucleation catalyzed in particular by surface hydrophobicity (either homogeneous or patchy) results in the lowering of the incipient boiling temperature.

The study is motivated by previous experimental investigations but clearly makes no claim to direct comparison with existing work in this subject area.

In fact, as the Referee suggests, monitoring surface roughness is intrinsically difficult, and we fully agree that the available techniques require specific care and case-by-case implementation.

However, from our results, we are led to believe that, at the nucleation level, the height of the local free energy barrier possessed by the (possibly heterogeneous) surface is the crucial element. From this point of view, the checkerboard, hydrophobic/hydrophilic configurations clearly show that the nucleation rate is determined only by the hydrophobic regions with lower activation energy. Such zones act as nucleation sites in a coarse-grained view of the liquid/vapor transition.

It can be speculated (even though for the moment we did not run specific simulations) that a similar mechanism works in the presence of sub-micrometrical roughness and/or dissolved gas nuclei. In fact, it is straightforward to make a reason for the former case, since already classical CNT (Blander, Milton, and Joseph L. Katz. "Bubble nucleation in liquids." *AICHE Journal* 1975) predicts a free-energy barrier decrease over geometric imperfections.

In the second case, also based on macroscopic experience and everyday life (as sparkling water nicely demonstrates), it is expected that dissolved gas lowers the nucleation barrier. As a matter of fact, the gas content modifies the spinodal line with respect to pure liquid decreasing the barrier, (Wang, Yuliang, et al. "Giant and explosive plasmonic bubbles by delayed nucleation." *Proceedings of the National Academy of Sciences* 2018).

From this more general perspective, the smooth wall can be seen as an extreme case to investigate the origin of the low onset temperatures that, according to our modeling, persists also in the case of hyper-smooth, ultra-pure, ideal liquids.

A promising configuration for the study of nucleation on atomically smooth surfaces has been explored by one of the authors (Pfeiffer, Patricia, et al. "Heterogeneous cavitation from atomically smooth liquid–liquid interfaces." *Nature Physics* 2022) in the context of heterogeneous nucleation of cavitation bubbles on liquid/liquid interfaces. To be precise, that paper, combining experiments and molecular dynamics, considered the presence of dissolved gas, but that configuration could inspire further subsequent studies at the mesoscopic level.

Finally, without any claim of direct comparison, it may be worth stressing that the typical size of our hydrophobic patches was inspired by the results reported by part of these authors in Ref. [6] where, by analyzing the surfaces by AFM and ellipsometry, cavities on the scale of 20-40 nanometers were detected.

Combining the above comments with the observation that we are probing scales that are hardly reached by physical experiment, we now provide better motivation to the work, as the Referee warmly recommends. We have amended the text accordingly in the new version of the manuscript.

MAJOR COMMENT #2

The results are presented in non-dimensional unit only. The authors discuss how to get back to dimensional units in the methods section. However, the reviewer thinks it would be appropriate to present some results in dimensional form to help the reader appreciate the significance of the work.

We thank the referee for bringing this important point to our attention.

In the original paper, dimensional quantities were reported in a scattered way by considering water as a working fluid. Following the Referee's suggestion, the new version of the manuscript systematically provides dimensional values, still using the properties of water as a reference.

We take this opportunity to further stress that, in general, the use of reduced variables (dimensionless with respect to critical state values) is convenient given their supposed universality. This allows us to reasonably extend to real fluids results obtained in the context of a Van der Waals model, suggesting that the observed behavior is generic for the nucleation process of fluids described by a mean-field approach.

Entering into more technical details, we also underline that for the sake of numerical convenience, simulations were performed at relatively high temperatures. In these conditions, the difference between the saturation and spinodal temperatures is small. However, the relevant reaction variable $\theta = (T - T_{sat}) / (T_{spin} - T_{sat})$ is expected to follow the same dynamics independent of temperature in the range between triple and critical points, as substantiated in a recent work on water cavitation by some of the present authors (Magaletti et al., *Scientific Reports* 2021), where the relevant variable was the reduced chemical potential.

Related to this comment, is also the observation that the heat fluxes used by the authors seem to be quite high. For instance, $Q = 0.001$ (which is the lowest non-dimensional heat flux) “corresponds to a few MW/m², the typical heat flux used in experiments.” In most experiments, onset of nucleate boiling occurs at fraction of a MW/m². In fact, the critical heat flux of water in pool boiling conditions is in the

order of 1 MW/m². Can the authors provide results for Q values corresponding to ~0.1 MW/m² (i.e., onset of boiling conditions)?

The reviewer's comment is most appropriate. We concede that the heat fluxes are high and that it is extremely interesting to investigate smaller ones ($Q=0.001$), as found in ordinary applications. Again, as in any practical case, we have to come to a compromise with resource limitations.

Roughly speaking, as the temperature rises at constant pressure, the fluid becomes progressively metastable with an increasingly reduced nucleation barrier. As a consequence, the nucleation time, exponential in the barrier height, goes from extremely large near saturation to vanishingly small as spinodal conditions are approached. The heat flux determines the rate at which the temperature raises. If the temperature raises slowly, even the large barrier at small temperatures can be crossed, since sufficient time is available. If the temperature raises fast, on the opposite, the temperature keeps rising before the barrier can be crossed, up to a point where the barrier becomes small enough such that nucleation takes place before the temperature can grow further. Hence, due to the exponential dependency on the barrier, the onset boiling time decreases at, approximately, an exponential rate by increasing the heat flux. This cartoon is consistent with the data that show an increase in the boiling onset time from 10^3 to 10^6 with a heat flux reduction from $Q=0.01$ to $Q = 0.001$. In conclusion, reducing the heat flux to $Q = 0.0001$, like in ordinary conditions, though conceptually feasible, is, in fact, extremely arduous computationally. Some of the present authors are starting to approach this problem through the more sophisticated tools of large deviation theory, but more work is still ahead before conclusive data are available.

Finally, we would like to stress that innovative technologies are emerging which imply heat fluxes on the order of those considered in the paper. Specifically, the ever-increasing power to be dissipated by modern microchips is already on the order of 5-10 MW per square meter, with peaks even tens of times the nominal values (Zhang, Zhihao, Xuehui Wang, and Yuying Yan. "A review of the state-of-the-art in electronic cooling." *e-Prime-Advances in Electrical Engineering, Electronics and Energy* 2021).

We have added a summary of this discussion in the new version of the manuscript.

MAJOR COMMENT #3

One may argue that when boiling has started, residues of vapor trapped on cavities or on the surface may serve as nucleation sites. In fact, experiments have shown that this nucleation sites can be identified (e.g., using infrared or other optical techniques) and are quite stable. Thus, the theory developed by the authors may apply to the onset of boiling, when there is potentially no micron-scale vapor (or non-condensable gas) structure.

The Referee is right. Certainly, a very common mechanism in the liquid/vapor phase transition consists of the mere accretion of pre-existing gaseous nuclei in the liquid. Honestly, we do not feel like speculating on what happens under real experimental conditions about the absorption of gas or vapor nuclei in wall asperities. In that case, we can only argue that the presence of electric charges, geometric discontinuities, and contaminating agents such as surfactants may play an important role. This observation does not claim any scientific rigor but is just a feeling we want to share with the Referee.

As far as our work is concerned, the experimentally observed nucleation sites are micrometrical objects which are well beyond the nucleation stage. In fact, the critical nuclei of pure liquids have a typical size in the range from a few to ten/twenty nanometers, depending on metastability degree. At those scales, pure vapor bubbles are strongly unstable, except for closed microcavities (Azouzi, Mouna El Mekki, et al., *Nature Physics* 2013). Actually, in open systems, only two thermodynamic stable conditions do exist, namely the homogeneous liquid and vapor, respectively.

The picture changes considerably in the presence of dissolved gas. This scenario can be tackled by extending the present approach to include a second species. Work in this direction is already available in the literature for deterministic approaches that do not incorporate nucleation (Liu et al., *Phys. Rev. E* 2016; Mukherjee et al. *Appl. Phys. Lett.* 2019; Benilov, *J. Fluid Mech.* 2023).

MINOR COMMENTS:

1. The abstract is very lengthy. I believe that Nature Communications sets a word limit on the abstract. I suggest the authors to simplify the abstract by focusing on the novelty and the impact of their work. The introductory part of the abstract discussing the motivation of the work can be shortened significantly, as it is detailed discussed in the introduction.

The abstract has been shortened accordingly.

2. The introduction is also quite lengthy. I advise the authors to write more concisely.

We did our best to re-modulate the introduction eliminating some redundancies.

3. The first section in the Results (i.e., how surface wettability influences boiling) can be structured in clearer way. For instance: It is not clear why the inset in Figure 2 is not presented with Figure 1; the definition of J_w is provided after its values are discussed in the text.

We have modified Fig.1 and Fig.2 and restructured the narrative accordingly.

4. There is an oversight in the intro that should be fixed, with respect to Ref. 18. The temporal resolution in ref 18 is ~1 millisecond, not microsecond. Infrared cameras are much slower than high-speed video camera. However, in Ref. 18, the authors measured the temperature at the back side of a silicon water (i.e., the side opposite to the one where boiling occurs). The authors could find better examples of high-speed video or high-speed infrared thermometry that focused on bubble nucleation.

We have corrected the typo. We will willingly accept suggestions on the most appropriate videos to mention.

5. How do you explain the fact that, in Figure 3, the temperature on the ultra-smooth wall goes below 0, i.e., below the saturation temperature?

Concerning Fig. 3, in order to isolate the nucleation event to a single bubble, we decreased the size of the system. When a hydrophobic patch is present, only a single bubble appears on the defect. Concurrent wall cooling inhibits bubble formation elsewhere, and just one bubble is left expanding on top of the hydrophobic region. When homogeneous walls are considered, the nucleation is delayed until it eventually takes place by producing bubbles scattered all over the wall. As a consequence of the larger amount of vapor and related latent heat absorption the temperature drops, leading the mean wall temperature to fall for a while below saturation. Afterward, the temperature raises again, and the familiar picture is restored.

In conclusion, we thank the Referee again for the work spent reviewing our paper. We believe that their suggestions allowed for an increased quality of the paper in terms of both content and exposition clarity. Based on the comments we received, we believe the paper can now be considered ready for publication in Nature Communications.

REVIEWERS' COMMENTS

Reviewer #1 (Remarks to the Author):

The authors addressed my comments. The manuscript is ready for publication.

Reviewer #2 (Remarks to the Author):

Dear Authors,

I appreciate your efforts in answering my questions and comments. My recommendation to the editor is to accept your revised paper.

Below are further comments that you may consider to further improve the quality of your work before the paper is published.

MAJOR COMMENTS:

MAJOR COMMENT #1:

The answer of the authors and the way the authors addressed this point in the manuscript is acceptable.

MAJOR COMMENT #2:

While I understand that running simulations at heat fluxes in the order of 0.1 to 1 MW.m⁻² is computationally prohibitive, I am more skeptical by the justification provided by the authors, i.e., that future micro-electronics devices will have to dissipate ~10 MW.m⁻². These heat fluxes are very hard to attain in nucleate boiling, even in subcooled and pressurized flow boiling conditions

(incidentally, experiments in Ref. [7] in subcooled flow boiling conditions at ambient pressure mentioned by the authors experienced a boiling crisis at $\sim 3.4 \text{ MW.m}^{-2}$ for the highest subcooling).

I feel the authors should confidently state why they did not run tests in more relevant conditions. As I believe other readers may rise the same question, I advise the authors to mention that in-silico experiments at lower heat fluxes, in the range of 0.1 to 1 MW.m^{-2} are computationally prohibitive and will be the object of future investigations with more sophisticated tools (as mentioned in their answer to my comment).

MAJOR COMMENT #3:

The answer of the authors is acceptable. As I believe other readers may rise the same question, I suggest the authors to add a comment (similar to their answer) in the paper.

MINOR COMMENTS

MINOR COMMENTS #1, #2, #3

I feel that both abstract, introduction and results section have been improved significantly.

One aspect that may still confuse the reader is the use of the term “macro” (e.g., “The model is able to describe boiling from the stochastic nucleation up to the macroscopic bubble dynamics”).

In my understanding, the work of the authors bridges the gap between nanometric and micrometric length scales but does not apply to the later stages of the bubble growth (from microns to millimeters). I advise the authors to revise the narrative to vet the use of the term “macro” vs. “micro”.

MINOR COMMENTS #4

As the authors are willing to accept suggestions on the literature to cite in support of their statement, they may consider “Jung, S. and Kim, H., 2014. An experimental method to simultaneously measure the dynamics and heat transfer associated with a single bubble during nucleate boiling on a horizontal surface. International Journal of Heat and Mass Transfer, 73, pp.365-375.”

MINOR COMMENTS #5

As I believe other readers may rise the same question, I suggest the authors to add a comment (similar to their answer) in the paper to explain such behavior.

Replay Referee # 1

NCOMMS-23-29819-A.

A nanoscale view of the origin of boiling and its dynamics. By Gallo et al.

The authors addressed my comments. The manuscript is ready for publication.

We thank again the Referee for the time spent reviewing our work and for the useful suggestion provided.

Replay Referee # 2

NCOMMS-23-29819-A.

A nanoscale view of the origin of boiling and its dynamics. By Gallo et al.

Dear Authors,

I appreciate your efforts in answering my questions and comments. My recommendation to the editor is to accept your revised paper.

Below are further comments that you may consider to further improve the quality of your work before the paper is published.

We thank the Referee again for the work spent on our manuscript and for the additional advice aimed at improving our work. As you can appreciate from the new version of the paper we have implemented all the additional suggested corrections, which are highlighted in blue for the Referee's convenience.

What follows are point-by-point answers to all the questions raised in the report.

MAJOR COMMENTS:

MAJOR COMMENT #1:

The answer of the authors and the way the authors addressed this point in the manuscript is acceptable.

Thanks.

MAJOR COMMENT #2:

While I understand that running simulations at heat fluxes in the order of 0.1 to 1 MW.m⁻² is computationally prohibitive, I am more skeptical by the justification provided by the authors, i.e., that future micro-electronics devices will have to dissipate ~10 MW.m⁻². These heat fluxes are very hard to attain in nucleate boiling, even in subcooled and pressurized flow boiling conditions (incidentally, experiments in Ref. [7] in subcooled flow boiling conditions at ambient pressure mentioned by the authors experienced a boiling crisis at ~3.4 MW.m⁻² for the highest subcooling).

I feel the authors should confidently state why they did not run tests in more relevant conditions. As I believe other readers may rise the same question, I advise the authors to mention that in-silico experiments at lower heat fluxes, in the range of 0.1 to 1 MW.m⁻² are computationally prohibitive and will be the object of future investigations with more sophisticated tools (as mentioned in their answer to my comment).

In the new version of the paper, we have modified the sentence as suggested and added a discussion about the difficulty of dealing with simulations at lower heat fluxes with a callback to the possible solution with rare-event techniques.

MAJOR COMMENT #3:

The answer of the authors is acceptable. As I believe other readers may rise the same question, I suggest the authors to add a comment (similar to their answer) in the paper.

As suggested, we have added a comment in the new version of the paper.

MINOR COMMENTS

MINOR COMMENTS #1, #2, #3

I feel that both abstract, introduction and results section have been improved significantly.

Thanks.

One aspect that may still confuse the reader is the use of the term “macro” (e.g., “The model is able to describe boiling from the stochastic nucleation up to the macroscopic bubble dynamics”).

In my understanding, the work of the authors bridges the gap between nanometric and micrometric length scales but does not apply to the later stages of the bubble growth (from microns to millimeters). I advise the authors to revise the narrative to vet the use of the term “macro” vs. “micro”.

Thank you for the clarification.

We believe the misunderstanding may arise from the specific use of the terminology of mesoscale physics in which physical quantities related to the granularity of matter are called microscopic. These quantities are inherited at the mesoscale as effective models, e.g., thermal noise, capillarity, and wettability. Conversely, macroscopic quantities are those inherent in the hydrodynamics of the system. To avoid misunderstanding, we added this clarification in the paper.

MINOR COMMENTS #4

As the authors are willing to accept suggestions on the literature to cite in support of their statement, they may consider “Jung, S. and Kim, H., 2014. An experimental method to simultaneously measure the dynamics and heat transfer associated with a single bubble during nucleate boiling on a horizontal surface. International Journal of Heat and Mass Transfer, 73, pp.365-375.”

Thank you for bringing this work to our attention. We have added it to the list of references and added a comment in the introduction.

MINOR COMMENTS #5

As I believe other readers may rise the same question, I suggest the authors to add a comment (similar to their answer) in the paper to explain such behavior.

As suggested, we have added a comment in the new version of the paper.